# Scale to Evaluate Employee Experience: Evidence of Validity and Reliability in Regular Basic Education Teachers in the Peruvian Context

**DOI:** 10.3390/bs14080667

**Published:** 2024-08-01

**Authors:** Nilton Acuña-Hurtado, Elizabeth Emperatriz García-Salirrosas, Miluska Villar-Guevara, Israel Fernández-Mallma

**Affiliations:** 1UPG de Ciencias Empresariales, Escuela de Posgrado, Universidad Peruana Unión, Lima 15102, Peru; nilton.acuna@upeu.edu.pe; 2Faculty of Management Science, Universidad Autónoma del Perú, Lima 15842, Peru; 3EP de Administración, Facultad de Ciencias Empresariales, Universidad Peruana Unión, Juliaca 21100, Peru; miluskavillar@upeu.edu.pe; 4EP de Ingeniería Civil, Facultad de Ingeniería y Arquitectura, Universidad Peruana Unión, Juliaca 21100, Peru; pastorisrael@upeu.edu.pe

**Keywords:** employee experience, validation, psychometric properties, dimensionality, Peru

## Abstract

Research on employee experience is a topic that has been growing in recent decades. This study analyzes the validity and reliability of an employee experience scale in Peruvian teachers. The study had an instrumental design. The sample was comprised of 760 Peruvian teachers between 20 and 71 years old (M = 40.91; SD = 10.05), where men (36.1%) and women (63.9%) participated, recruited through non-probabilistic sampling. A validity and reliability analysis of the employee experience scale confirmed the three original factors (sensory experience, intellectual experience, and emotional experience). The KMO test reaches a high level (0.950 > 0.70), and the Bartlett test reaches a highly significant level (Sig. = 0.000). The scale also showed good internal consistency (α = 0.948 to 0.980; CR = 0.950 to 0.981; AVE = 0.864 to 0.878). Similarly, for the confirmatory factor analysis, a measurement adjustment was performed, obtaining excellent and acceptable fit indices for Model 2 for three factors (CMIN/DF = 4.764; CFI = 0.984; SRMR = 0.024; RMSEA = 0.070). This study provides a useful tool to measure the employee experience in a friendly way, using simple language to be applied to the Peruvian context. This study is considered an important contribution to organizational behavior and human talent management in educational circles.

## 1. Introduction

The relationship between employer and employee has been undergoing important changes in recent years [1,2,3]. The book *Employee Experience Advantage* by Morgan [4] describes this evolution in four stages, which will be explained in the following paragraphs. The usefulness stage was where the relationship between both was based on the employer providing the tools or equipment that were useful for the employee to fulfill his/her function (pencil, desk, laptop, smartphone). In this stage, the priority was to fulfill the responsibility given to the worker. Following this prior stage came the productivity stage, in which the relationship focused on measuring how much a worker could produce. The center was what the worker produced, not the worker himself [5].

The commitment stage came later as a new concept, where the worker’s collaboration was transcendental in the company [6]. This approach is important, since the employer understands that the organization can benefit when it cares about its employees and knows their aspirations and motivations [4,5]. However, the commitment came to improve relationships only in the short term, since the changes are only temporary and superficial, due to the fact that these actions make things look better, but have little impact on the employee’s actual performance [5]. In this sense, the employee experience came as the long-term redesign of the organization, where not only the results of the workers are sought, but also that the employee can satisfy their needs and desires [6].

A recent qualitative study reported its findings on employees’ experiences of a five-day strategic program where teams experienced a deep sense of accomplishment, shared their successes, and developed positive attitudes toward teamwork [7]. On the other hand, a recent theoretical contribution analyzes employees’ experiences of the challenges and opportunities that people face and the problems that employers face creating new ways for organizations to handle challenging scenarios. Given the managerial proposals provided by this study, they suggest that future research can also explore other environments, such as the educational context [8]. Patil et al. [9] conducted a study on 201 academic workers, where they argue that employee experience provides a holistic and motivational view for performance at all levels of the organization. Employees who have a sense of belonging, purpose, achievement, energy, and happiness are more likely to perform at higher levels, and institutions that invest heavily in the employee experience are rated as the best places to work and achieve positive results.

After a few decades of the evolution of this topic, the interest in its application to educational environments has become evident [9,10,11,12,13,14,15,16], and at the same time, scientific studies have shown that employees who feel valued and satisfied are more likely to be productive and focused at work [17]. Considering this, evaluating employee experiences can reveal areas that can be improved to increase satisfaction [18] and work commitment [19] and thus increase productivity [20]. Additionally, it helps to identify workplace issues such as harassment, discrimination, and miscommunication [11,15,21]. Addressing these issues will greatly improve the work environment and promote a culture of respect and cooperation [22,23,24]. Knowing the experience of employees will help identify which aspects of the organizational culture, work environment, and organizational policies are contributing to improving the perception of each worker. This is critical for the long-term success of an organization [16,25].

Measuring the experience of employees in educational environments is important because it provides benefits in various aspects [26,27], such as talent retention [28], productivity [29,30], spirituality and emotional intelligence [31], commitment [32,33], a better work environment [34,35], and in creativity and innovation [36,37,38]. Improving the employee experience becomes important for creating a productive and trained workforce that contributes to the success and stability of society [39].

The reason why organizations strive to improve the employee experience is because it positively affects job satisfaction, commitment and retention, performance and productivity, attitude, and the happiness of employees [40,41,42,43], which are considered important internal factors in every organization. However, the benefits are not only limited to an internal scenario, but can also have an important impact on customer experience and satisfaction, and consequently, on profitability, brand reputation, productivity, the attraction of other talents, and the organizational culture [44,45]. In view of this, there is interest in studying this construct in greater depth, since there is no metric that evaluates the perception of Peruvian teachers. Regarding this, it is believed that providing the scientific community and educational leaders with a valid instrument that can evaluate the experience of employees, in the context of Regular Basic Education (RBE) institutions in Peru, fills the knowledge gap and offers a great contribution to the theoretical advancement of this construct. For this reason, the objective of this research is to adapt and evaluate the validity and reliability of an employee experience scale to Peruvian teachers of a network of private educational institutions in Peru.

## 2. Literature Review

### 2.1. Employee Experience

For context, it is necessary to remember an important quotation by Jacob Morgan [6], where he says, “In a world where money is no longer the primary motivating factor for employees, focusing on the employee experience is the most promising competitive advantage that organizations can create” (p. 84). A clear way to define employee experience is to consider it as a comprehensive concept that encompasses all the interactions and perceptions of an employee throughout his or her life cycle within an organization, starting from recruitment through onboarding and exit. This term goes beyond simple job satisfaction and includes aspects that affect how employees perceive their workplace and their role in an institution.

Studies argue that employee experience is more about cultivating and sustaining strong and productive engagement through the day-to-day experiences of an organization’s existing workers [46,47,48]. Employee experience is the most important performance indicator in the world of human resources [49]. It is a key factor for improving technical performance [50]. It is recognized as one of the motivators with the greatest impact of human capital in organizations [17,51,52]. For this reason, companies that invest in improving the employee experience are more likely to be considered a “good place to work” and attract better candidates to their organization [53].

Some studies claim that organizational environments are often characterized by constant tension; however, many managers ignore this [54]. That is, human talent management specialists continue to study the organizational factors that affect the employee experience, [55,56] while employees face various challenges in their workplace on a daily basis [57,58] that cause them to suffer psychological effects such as excessive stress [59], mental crisis, depression, etc. [60,61,62]. In spite of that, others have no choice but to adapt [63]. The employee experience is a strategy that recognizes employees as key stakeholders in an entity and key influencers in the external reputation and internal culture of the same [48]. Employees with positive experiences tend to be well connected to the organization [64]; they are more productive [65] and responsive [6,66]. In addition, studies focused on educational settings argue that a positive employee experience continues to be a valuable response to strengthen connections on emotional, social, physical, and spiritual levels [9,10,11,12,13,14,15,16].

Because employee experience is a concept that continues to evolve [18,20], it has been deemed necessary to make a diligent review of the positions of some specialists. One of the latest proposals that has generated an impact in the scientific community is the one that refers to the employee experience as the common path that employees take when interacting with an organization, based on common ground between the wants, needs, and desires of the employees and the organization itself [32]. Researchers with a qualitative approach have a special interest in the study of the employee experience, making their contributions bring greater depth to this construct [49,50,53,54,63,67,68,69]. Below is a review of definitions of the employee experience (Table 1):

### 2.2. Scales to Evaluate Employee Experience

When reviewing the diversity of proposed definitions, important scientific studies have been found which maintain that the employee is usually associated with organizational commitment [32], team coaching [53], leadership behaviors [73], systematic management [74], overall performance [16], customer loyalty, perceived authenticity and relational commitment [33], brand authenticity [48], employee engagement [75], loyalty and brand experience [76], innovative behavior [77], reduction of burnout, turnover, and the generation of well-being [59].

Most of the published content on this construct has been in book format and trade (non-scientific) journals, and to the present date there have been very few scholarly contributions from primary sources [48,66]. It is estimated that the reason for this shortage may be due to the lack of metrics to help assess employees’ perception of their workplace. Additionally, the review of previous studies supports the importance of having valid instruments that can measure the employee experience in various sectors of industry. For this, these instruments must have valid psychometric properties to be applied to different realities. However, it is necessary to mention that the employee experience is one of the constructs with scarce scientific support for studies with a quantitative approach in Peru and in various places around the world. In the following paragraph is a review of the measurement scales published in high-impact journals:

In India, Yadav and Vihari [66] constructed the EX Scale, which consists of 61 items. This scale was aimed at multinational cooperation workers and has six dimensions: (1) cohesiveness, (2) vigor, (3) well-being, (4) achievement, (5) inclusiveness, and (6) physical environment. The instrument has a Cronbach’s Alpha between 0.812 and 0.927. A seven-point Likert-type response scale was used, ranging from 1 = strongly disagree to 7 = strongly agree.

In the USA, Morgan [6] built a scale to measure the employee experience which consists of 10 items. This scale was aimed at workers from national companies and has three dimensions: (1) Physical Experience (example item: “I am proud to bring guests such as friends or family to the office”), (2) Technical Experience (example item: The “Our company systems are easy to use and useful”), and (3) Cultural Experience (example item: “Our company encourages diversity and inclusion”). The instrument has a Cronbach’s Alpha between 0.813 and 0.864.

In India, Patil et al. [9] developed and validated a multidimensional scale to measure employee experience in 21 items. This scale was aimed at academic workers and had four dimensions: leadership, HR practices, culture, and company image. The instrument has a Cronbach’s Alpha of 0.963. A seven-point Likert-type response scale ranging from 1 (strongly disagree) to 5 (strongly agree) was used.

In India, Pandita and Kiran [16] constructed a short unidimensional scale that assessed the employee experience as a construct in two dimensions: (1) employee attraction and (2) employee involvement. It was considered that the employee experience encompasses the involvement and attraction of employees, and that this directly influences research and its quality, as well as its reputation. They argued that employability refers to meeting the expectations of employers and the reputation of the employer. The scale was originally applied to university teachers. The instrument has a Cronbach’s Alpha of 0.812, a CR of 0.914, and an AVE of 0.842.

## 3. Materials and Methods

### 3.1. Study Design

An instrumental and cross-sectional study was conducted to examine the validity and reliability of an employee experience measurement instrument [78].

### 3.2. Sample and Procedures

For the present study, the population was made up of 1644 EBR teachers from preschool, elementary, and high school levels from a private educational network in Peru. This educational network bases its differentiating model and educational philosophy on the integral education of students (the harmonious development of physical, mental, and spiritual faculties), this being the most important pillar of this association of schools, which makes their teachers have very particular characteristics and which over time has been the focus of interest of academics in Peru and many countries in the world.

The conditions for employees to participate in this study were that they had to be classroom teachers who were not performing administrative activities and that had worked for the educational network in the period of 2023. This study had the approval of the ethics committee of the Graduate School of a private university in Peru (2024-CE-EPG-00027) and the authorization of the administration of each institution throughout the educational association. The virtual survey was hosted through Google forms and shared via WhatsApp, through the directors of each educational institution. All teachers in the educational network (74 colleges and schools) were invited, and a total of 760 teachers responded, between 20 and 71 years old (M = 40.91, SD = 10.05), selected through non-probability convenience sampling [79]. At the beginning of the survey, there were instructions indicating that the participation had to be of a voluntary nature and that the survey would be completely anonymous; if they agreed, they would give informed consent to proceed with the survey, thus following all the ethical principles for research with human beings as defined in the Declaration of Helsinki [80,81]. The application of this study was during the first semester of 2024. Table 2 shows the sociodemographic characteristics of the sample.

### 3.3. Instrument

The model proposed by Gavilan et al. [82] was used, which measured employee experience through 3 articulated dimensions—sensory experience (based on the workspace), intellectual experience (focused on values), and emotional experience (oriented to the enjoyment of work)—adapted from the literature and the opinion of specialists in human talent management. This metric was selected because its length is short enough to avoid bias, the Cronbach’s Alpha values indicated a high internal consistency, the items are culturally appropriate, and it was clear and easily understood by teachers, avoiding ambiguities. Additionally, it was appropriate to be applicable to a teaching community, filling the shortage of metrics in this demanding environment. Gavilan et al. [82] uses 4 items to measure the sensory experience, adapted from the brand experience scale of Brakus et al. [76]. To measure intellectual experience, it uses 7 items based on Berthon et al. [83]. Finally, to measure the emotional experience, it uses 3 items that come from the WOLF scale of Bakker [84]. The constructs were coded as follows: sensory experience (SE), intellectual experience (IE), and emotional experience (EE).

The questionnaire consisted of 14 items (Table 3). The questionnaire was prepared in the following way: The first section explained what the questionnaire consisted of, the instructions for filling it out, and the informed consent that the participants had to accept if they were willing to participate. The second and third sections had the scale items. To answer these items, the 5-point Likert scale was used, where 1 is totally disagree and 5 is totally agree. In the fourth section, the participant had to fill out the sociodemographic data.

### 3.4. Data Analysis Procedure

Two types of statistical software were used to analyze the data: (1) SPSS 25 was used to evaluate the exploratory factor analysis. (2) To perform the confirmatory factor analysis and evaluate the convergent and discriminant reliability and the fit of the measurement model, a covariance structural equation model (CB-SEM) was used, for which AMOS 24 software was used [85].

## 4. Results

Table 4 presents the descriptive statistical results of the items, such as the mean, standard deviation, skewness, and kurtosis of the scale. It is observed that the skewness and kurtosis values are mostly less than ±1.5 [86], except for items IE5, IE6, and EE1 of the kurtosis column, which showed a slight non-compliance with normality multivariate data. The maximum likelihood method was used, because it has the advantage of producing estimates that are asymptotically efficient and consistent, and with large samples, the estimate is robust to a slight violation of the multivariate method assumption of non-normality [87].

### 4.1. Exploratory Factor Analysis

To carry out the exploratory factor analysis (EFA), data from 360 respondents aged between 21 and 65 years (M = 40.77; SD = 10.45), 230 women (63.9%) and 130 men (36.1%), were used. The result of this analysis showed that the items are distributed into three factors depending on the construct examined (Table 5). The difference is quite clear between the factors. The KMO and Bartlett test (Kaiser–Meyer–Olkin correlation coefficient = 0.950) had a value greater than 0.70, and the Bartlett test (Sig. = 0.000) was very significant for carrying out a factor analysis. The total variance explained in the model was 88.244%, which is greater than 50%, being Intellectual Experience (IE) = 74.752%, Sensory Experience (SE) = 10.512%, and Emotional Experience (EE) = 2.960%. Subsequently, a confirmatory factor analysis (CFA) was carried out.

### 4.2. Confirmatory Factor Analysis

The confirmatory factor analysis was carried out with data from 500 participants (100 were also used for the exploratory factor analysis and 400 were a completely different sample) aged 20–71 years (M = 41.36; SD = 9.89); in this case, there were 316 women (63.2%) and 184 men (36.8%).

The validation of the final measurement model with convergent reliability and validity is evident in Table 6. Cronbach’s Alpha (α) values were between 0.948 and 0.980, considered satisfactory values, since all levels of this coefficient must be above 0.70 for the model to be valid [88]. Furthermore, the reliability values (CR) were between 0.950 and 0.981, which is favorable, because this value must be greater than 0.70 to be considered a perfect model [89]. Likewise, the AVE showed values between 0.864 and 0.878, which are considered acceptable, since this index must be equal to or greater than 0.50 [90]. In that sense, these values translate as an acceptable measurement model that meets favorable levels of reliability and convergent validity.

Figure 1 demonstrates the factor structure of the employee experience scale in a sample of 500 teachers from a private educational network in Peru.

What is shown in Table 7 are the goodness-of-fit indicators of the measurement model of the employee experience scale. The reported findings from the CFA of a three-dimensional model showed that the 14 items of the scale represented these three factors (Model 1). However, not all goodness-of-fit indicators were excellent; for this reason, the model was respecified based on the modification index (MI) [91]. In that sense, due to the similar wording of the items, there were correlations between the errors of some of them. In this way, the measurement model was analyzed by eliminating item IE7 and correlating the errors as follows, e6 with e7 and e8 with e11 (Model 2), obtaining excellent and acceptable fit indices.

The Fornell–Larcker criterion [85] was used to evaluate the discriminant validity of the model; for each factor, the square root of the AVE was calculated, which must be greater than the highest correlation between the factors in the measurement model [90]. Table 8 shows that all the values in the bold diagonal were greater than the correlation. Furthermore, Heterotrait–Monotrait (HTMT) criteria were also considered in this study [91]. If the HTMT value is less than 0.90, it is considered that there is discriminant validity between two reflective constructs. In this sense, it was also observed that the highest correlation had a value of 0.850, which is less than 0.90. With these results, the discriminant validity of the model is met.

## 5. Discussion

The aim of this study was to adapt and analyze the validity and reliability of an employee experience scale, a model constructed by Gavilan et al. [82]. Because this scale has not been validated in the Peruvian context and because of the need to review its relevance, this study was conducted with a population of 760 RBE teachers, a larger sample than that used in the original scale. The scale evaluates three factors distributed in 13 items. However, other scales such as that of Patil et al. [9] have been constructed from four factors, distributed in 21 items. Due to the scarcity of validated employee experience scales in the educational context, comparisons with other scales may be limited. The instruments used to measure employee experience have focused mainly on organizations in various business sectors, and whose factors do not necessarily focus on a worker’s experience in the educational sector. Thus, when reviewing the factors used in the preceding scales, some focus on positive organizational practices, stress level, well-being at work, and social well-being [92]. However, this does not mean that these factors cannot be studied in educational contexts. On the other hand, it is observed that there is no specific pattern that harmonizes in terms of the factors used, which represents a challenge when constructing a scale [7,66].

Another particular case is the scale designed and validated by Patil et al. [9] in a sample of 400 academic workers, which addresses the employee experience using 21 items out of 29 initially formulated. When comparing the procedures adopted for the validation of both scales, it is observed that in the present study, a convenience sampling was performed, while Patil et al. [9] used a two-step cluster sampling. Regarding the reliability test, which allows the measuring of the internal consistency of the metric, the EFA showed a grouping of the items similar to the original scale (in three factors), and in the CFA, the Cronbach’s Alpha values were between 0.948 and 0.980, being slightly higher (between 0.947 and 0.961) than that of the Gavilan et al. [82] scale. The CR values presented a slight variation between the results of this study (between 0.950 and 0.981) and those reported by Gavilan et al. [82] (between 0.966 and 0.968). On the other hand, the AVE indicators of this research ranged between 0.864 and 0.878, while Gavilan et al. [82] reported lower scores than these (between 0.812 and 0.905). The scale constructed by Patil et al. [9] grouped the variable into four factors (leadership, human resources practices, culture, and company image).

While this scale has been validated on the basis of three factors (sensory experience, intellectual experience, and emotional experience), other scales have included factors such as cohesion, vigor, well-being, achievement, inclusion, and physical environment. In addition to technical experience, physical experience, and cultural experience [6,16,66,93], unidimensional scales of this construct have also been constructed [16,93]. When comparing the reliability of the scales, this one stands out for its higher scores in terms of internal consistency. Although employee experience continues to be of interest in various fields of application, scientific production in the educational field is very limited; even more so, if metrics are needed to assess this construct focused on educational settings. Empirical contributions in the sector of education have suggested a connection with well-being [7], spirituality [94], burnout reduction [59], academic culture [16], intellectual disability [14], workplace violence [95], positive practices [92], and moral dilemmas [96]; although few studies have reported the psychometric aspects of the measurement scales used, they have revealed the need to generate instrumental contributions.

Since the employee experience has not been studied in the Peruvian context, this research represents a valuable contribution to deepen this field of study, given that few studies have been conducted in the educational context, which represents a challenge and an opportunity for researchers [94,95]. The reality of Latin America and Peru regarding employee experience reveals serious shortcomings due to working conditions and the commitment that employees show towards their organizations. In the educational field, this implies high teacher turnover and la ow commitment to the organization, which reveals a low level of satisfaction [97]. Addressing this reality remains a huge challenge for the Peruvian government, as well as for private entities in the country.

### 5.1. Theoretical and Practical Implications

According to Caplan [98], organizations should focus on three main objectives: retention, engagement, and innovation. In relation to this point, very often the work of human talent management focuses on the objectives and needs of the company, seeking to influence the behavior of employees to achieve those objectives, regardless of whether they are consistent with the basic needs (psychological and social) of those employees [71]. This study evaluates the psychometric properties of a scale of employee experience using three dimensions, expanding the research bases on this construct. This study enriches the literature by analyzing the conceptual definitions used in different studies, making a great contribution to future research.

For business professionals, empirical validation of the three dimensions of employee experience provides detailed knowledge on its strategic use to promote beneficial employee behavior in an organization. This study provides a list of variables valued by employees, which, if followed correctly, will produce results that benefit all employees, regardless of their position. Business professionals can also apply this learning by focusing on this construct to create a unique employee experience. And in turn, this instrumental research contributes to the understanding of the individual and organizational factors that influence the employee experience in a Peruvian context. Their findings can be used when implementing improvement programs for talent retention and engagement, within educational companies that need to increase their efficiency and effectiveness. For example, at the corporate level, these results suggest that the implementation of programs that target effective communication, strong organizational culture, and a positive work environment could be effective in the educational context.

At the management level, each proposal that aims to optimize the employee experience must encourage and strengthen their motivation to change their behavior [99,100]. For example, providing wellness and emotional support resources to help employees manage anxiety, stress, and other personal issues [100,101]. This may include counseling, wellness programs, and mental health policies, as focusing on educational efforts to ensure employees understand that the benefits of each program contribute to their own well-being comes with many benefits [26,100,101].

Regarding the way in which the work–life balance of educational employees is being addressed, it is necessary to take a look from the employee’s point of view, since on many occasions, this could be causing them to not have a good perception of their employer. Faced with this, he/she could offer flexibility in schedules, teleworking when possible, recognition with family time for personal achievements, a more empathetic disposition in the face of family misfortunes that may affect the employee’s work life, and the design of leave policies that allow employees to attend to their personal responsibilities, all of which could have important implications at the business level.

Many private educational institutions in Peru need to design an optimal salary schedule for teachers—that is, a base salary to attract and retain effective teachers—since this can be an elementary factor in the employee experience. Looked at from the perspective of human talent development, ensuring an equitable and fair salary, and in addition to this, offering market-competitive benefits and compensation that are attractive to employees, can improve their experience in the organization. In this sense, policy makers and salary scales in the sector must be consistent with the literature that supports the inequality between teaching contribution and salary compensation. It also highlights the importance of conducting more research on the workforce economics of the education sector and generating more scientific evidence on the effectiveness of policies to improve the experience of its employees.

This study suggests that the evaluation of digital and technological human talent systems (such as gamification and artificial intelligence) can influence the employee experience. These changes will surely provide a better work experience for employees. In addition, the effective use of these technologies can help educational leaders measure and improve behaviors that would otherwise be difficult to achieve. Artificial intelligence can collect data and provide useful information to improve the employee experience. Finally, this study increases our understanding of how Peruvian educators perceive the experience in their workplaces and what factors contribute to their success, measured from Sensory Experience, Intellectual Experience, and Emotional Experience. These results can help educational managers make more informed strategic decisions and improve their organizational actions and their strategic, tactical, and annual plans.

### 5.2. Limitations and Future Research

This research also has some limitations. First, it used cross-sectional data that could create potential biases that should be considered in future studies. Observing the employee experience from different time periods could be an important contribution to making more informed management decisions. Another limitation is due to the fact that the sample of this study comes only from Peru, which clarifies the life, experiences, perceptions, and working conditions of workers from this country. In this sense, the results of this study cannot be generalized to other countries and/or cultures. A future study should test the scale of employee experience in other Latin American contexts and various cultures around the world, thus increasing its generalization.

This study evaluated data collected from Peruvian RBE teachers affiliated with private institutions. Since public management differs in several aspects from the private sector, it is considered that it would be a great contribution to the academic community to analyze the results of the behavior of this construct in comparative or differentiating analyses. Future research could also focus on examining sectoral and sociodemographic differences in the same set of employee experience factors, to gain a deeper understanding of this construct. Another limitation that should be considered in the future is that this study did not take into account the proportion of participating teachers, their gender, or their level of academic instruction, which could generate a possible bias, and in light of this, it is suggested that future studies consider an equitable proportion of the sample.

Although this study only considered a teaching sample, it is believed that future research could evaluate the psychometric properties of the employee experience scale where non-teaching staff are also included. This is in order to analyze the behavior of this construct, since its impact transcends the entire organization in general. Finally, it is recommended that future empirical research include other associated factors that could impact the behavior of the employee experience, such as those referred to in previous studies: performance evaluation [64], organizational commitment [32], potential risk of burnout [59], innovative behavior [77], employee well-being [56] or other associated variables.

## 6. Conclusions

The main objective of this study was to evaluate whether the psychometric properties of a scale of employee experience was totally reliable, flexible, and practical to be applied by professionals and academics of organizational behavior and human talent management. In that sense, having a scale for measuring the employee experience is essential, as it provides a valid tool to evaluate and improve the work environment. In addition, it can be very useful for identifying work problems, making informed decisions, monitoring over time, improving talent retention, and promoting organizational commitment. It seeks to become an ally for the positioning and reputation of the brand. The scale of employee experience can be very useful in facilitating the creation of a coherent and positive work environment.

A validity and reliability analysis of the scale of employee experience confirmed the three original factors (Sensorial Experience, Intellectual Experience, and Emotional Experience). The KMO test reaches a high level (0.950 > 0.70), and the Bartlett test reaches a highly significant level (Sig. = 0.000). The scale also showed good internal consistency (α = 0.948 to 0.980; CR = 0.950 to 0.981; AVE = 0.864 to 0.878). Similarly, for the confirmatory factor analysis, a measurement adjustment was performed, obtaining excellent and acceptable fit indices for Model 2 for three factors (CMIN/DF = 4.764; CFI = 0.984; SRMR = 0.024; RMSEA = 0.070). This study provides a useful tool to measure the employee experience in a friendly way, using simple language to be applied to the Peruvian context. This study can be considered an important contribution to organizational behavior and human talent management in educational circles.

## Figures and Tables

**Figure 1 behavsci-14-00667-f001:**
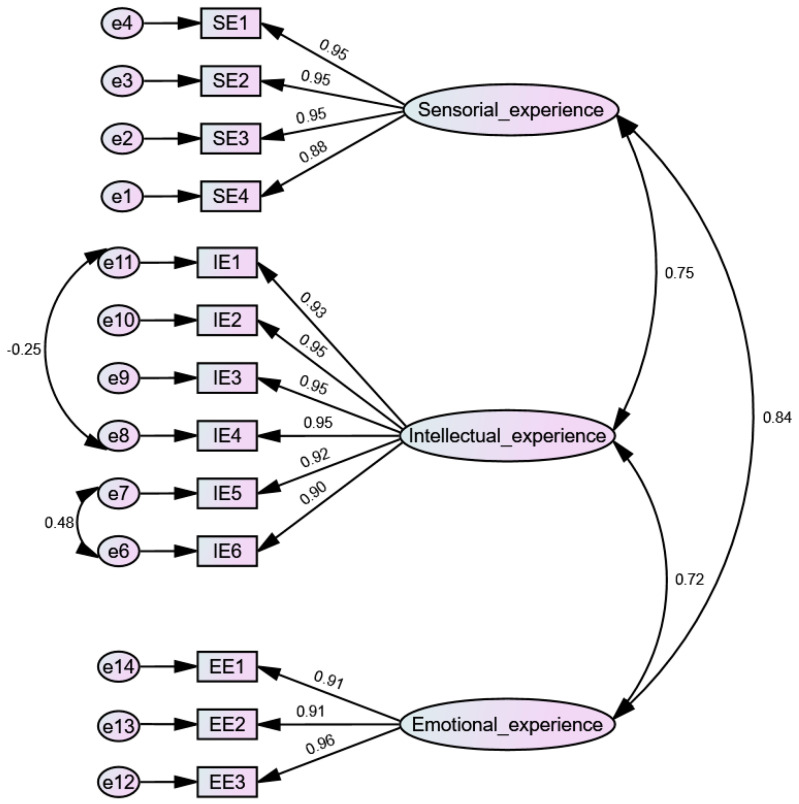
Measurement model of gratitude organization in the Peruvian context.

**Table 1 behavsci-14-00667-t001:** Definitions of employee experience.

No.	Definition of Employee Experience	Fountain
1	The set of opinions or points of view acquired through various interactions during an employee’s stay in the organization, from the initial contact with the potential employee until his or her separation from work.	Yadav and Vihari [66]
2	“It is the sum of every employee interaction, from the first contact with a potential employee to the last interaction after the employment relationship ends. It extends beyond traditional HR functions to include facilities, corporate communications, risk and compliance, IT, and more” (p. 34).	Yohn [48]
3	It is a shift towards a co-creation perspective based on the importance of employee engagement experiences, designed to meet employee needs and expectations.	Lemon [69]
4	It is the subjective perception that the employee has of his or her work experience while working in the organization.	Shenoy and Uchil [70]
5	It is known as an individual’s general vision of their relationship with the organization in which they work. All meeting at contact points throughout their work journey.	Plaskoff [71]
6	It is the sum of cognitive, behavioral, sensory, emotional, “and social responses that occur in interactions with other parties” (p. 240).	Larivière et al. [72]
7	“It is the intersection of employee expectations, needs, and wants and the organization’s design to meet employee expectations, needs, and wants” (p. 91).	Morgan [6]
8	It involves all aspects of work life, from the work environment itself and relations with management to personal efforts.	Patil et al. [9]

**Table 2 behavsci-14-00667-t002:** Sociodemographic characteristics of the participants (*n* = 760).

Characteristic	Category	Frequency	Percentage (%)
Sex	Female	486	63.9
	Male	274	36.1
Age range	20–29 years	125	16.4
	30–39 years	191	25.1
	40–49 years	281	37.0
	50–71 years	163	21.4
Academic level	Technical	234	30.8
	Bachelor	342	45.0
	Master	181	23.8
	Doctor	3	0.4
Geographic location	Coast	410	53.9
	Mountain range	190	25.0
	Jungle	160	21.1

**Table 3 behavsci-14-00667-t003:** Description of the construct in English and Spanish.

Predictor	Items	Description
		*En mi institución educativa, el lugar o espacio donde trabajo... [In my educational institution, the place or space where I work...]*
Sensory Experience (SE)	SE1	Es agradable para trabajar [*Is pleasant to work in*]
	SE2	Me gusta [*I like it*]
	SE3	Hace que me sienta bien [*Makes me feel good*]
	SE4	Me facilita hacer bien mi trabajo [*Makes it easier for me to do my job well*]
		*Los valores corporativos de mi institución educativa... [The corporate values of my educational institution...]*
Intellectual Experience (IE)	IE1	Los conozco [*I know them*]
	IE2	Los comprendo [*I understand them*]
	IE3	Los comparto [*I share them*]
	IE4	Me identifico [*I identify with them*]
	IE5	Son positivos para la sociedad [*Are positive for the society*]
	IE6	Son positivos para los empleados [*Are positive for the employees*]
	IE7 *	Son positivos para los alumnos y padres de familia [*Are positive for the students and parents*]
		*Trabajando en mi institución educativa... [Working in my educational institution...]*
Emotional Experience (EE)	EE1	Me siento bien [*Makes me feel good*]
	EE2	Me divierto [*I have fun*]
	EE3	Disfruto [*I enjoy myself*]

Note: * Item IE7 was excluded in the confirmatory analysis for a better fit of the measurement model.

**Table 4 behavsci-14-00667-t004:** Descriptive analysis of the items (*n* = 760).

Code	Mean	Standard Deviation	Skewness	Kurtosis
SE1	4.0750	1.02069	−1.090	0.834
SE2	4.1211	1.01430	−1.133	0.863
SE3	4.0592	1.01526	−1.005	0.569
SE4	3.9803	1.04238	−0.961	0.473
IE1	4.1684	0.94544	−1.120	1.015
IE2	4.1711	0.95606	−1.118	0.930
IE3	4.1803	0.96466	−1.153	0.953
IE4	4.2276	0.95458	−1.279	1.359
IE5	4.3026	0.92427	−1.368	1.598
IE6	4.2579	0.95308	−1.331	1.516
IE7	4.2684	0.95156	−1.340	1.497
EE1	4.2553	0.94547	−1.420	1.945
EE2	4.1053	0.97097	−1.069	0.921
EE3	4.1763	0.96334	−1.183	1.128

**Table 5 behavsci-14-00667-t005:** Exploratory factor analysis (EFA) pattern matrix: own elaboration.

	Factor		
	1	2	3
IE7	0.955		
IE5	0.955		
IE3	0.935		
IE6	0.928		
IE4	0.911		
IE2	0.880		
IE1	0.849		
SE2		0.989	
SE1		0.941	
SE3		0.901	
SE4		0.794	
EE3			0.885
EE2			0.806
EE1			0.523

Extraction method: maximum likelihood. Rotation method: Promax with Kaiser normalization.

**Table 6 behavsci-14-00667-t006:** Validation of the final measurement model with convergent reliability and validity.

Predictor	Items	Estimate	α	CR	AVE
Sensory Experience (SE)	SE1	0.948 ***	0.963	0.964	0.869
	SE2	0.947 ***			
	SE3	0.947 ***			
	SE4	0.885 ***			
Intellectual Experience (IE)	IE1	0.906 ***	0.980	0.981	0.878
	IE2	0.942 ***			
	IE3	0.934 ***			
	IE4	0.939 ***			
	IE5	0.946 ***			
	IE6	0.936 ***			
	IE7 *	0.955 ***			
Emotional Experience (EE)	EE1	0.911 ***	0.948	0.950	0.864
	EE2	0.914 ***			
	EE3	0.962 ***			

Cronbach’s Alpha (α) for all variables is >0.70, the composite reliability (CR) > 0.70, and the mean variance extracted (AVE) > 0.50; *** *p* < 0.001 (significance level), indicating the significant validity of the model. * Item IE7 was excluded in the confirmatory analysis for a better fit of the measurement model.

**Table 7 behavsci-14-00667-t007:** Statistical goodness-of-fit indices of the employee experience scale.

Measure	Threshold	Model 1		Model 2	
		Estimate	Interpretation	Estimate	Interpretation
CMIN	--	669.954	--	285.868	--
DF	--	74.000	--	60.000	--
CMIN/DF	Between 1 and 3	9.053	Terrible	4.764	Acceptable
CFI	>0.95	0.963	Excellent	0.984	Excellent
SRMR	<0.08	0.023	Excellent	0.024	Excellent
RMSEA	<0.06	0.103	Terrible	0.070	Acceptable

Note: CMIN = chi square, DF = degrees of freedom, SRMR = standardized root mean square residual, RMSEA = Root Mean Square Error of Approximation, CFI = comparative fit index. Model 2: Item IE7 excluded, e6–e7; e8–e11.

**Table 8 behavsci-14-00667-t008:** Discriminant validity.

Fornell–Larcker Criterion				Heterotrait–Monotrait Ratio (HTMT)	
	SE	IE	EE	Correlation	HTML
SE	**0.** **932**			IE-SE	0.747
IE	0.745 ***	**0.** **933**		EE-SE	0.850
EE	0.838 ***	0.719 ***	**0.** **930**	EE-IE	0.731

Note: The square root of AVEs is shown diagonally in bold, *** *p* < 0.001 (significance level).

## Data Availability

Data can be requested by writing to the corresponding author of this publication.

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
