# Peer review of "Scale to Evaluate Employee Experience: Evidence of Validity and Reliability in Regular Basic Education Teachers in the Peruvian Context"

_behavsci, 2024, doi:10.3390/bs14080667_

Round 1

Reviewer 1 Report

Comments and Suggestions for Authors

I would like to thank for the opportunity to review this manuscript. The topic is interesting and important, as the availability of appropriate scales for gathering information about employee experience in an organisation is crucial from the aspect of improving the quality of work environments and increasing employee job satisfaction. There are several key areas that need further work before publication. I have summarised my recommendations, questions, and the required changes in the hope that the feedback will be useful to you.

1)     Introduction and Literature Review – They are poorly written and do not refer adequately to previously published studies. More empirical data should be added, so the authors need to carefully review recently published studies and also pay attention to the specifics of school environments rather than operating with business entities. Undoubtedly – even if they are private institutions – there are significant differences between schools and other types of organisation, and also the factors that affect the ‘employee experience’ of teachers and job (dis)satisfaction are different. I suggest rewriting these two sections and including information on 1. school environments where teachers gain their employee experience and 2. their job satisfaction. In this form, the content does not correspond to the purpose of the study.

2)     Regarding the methodology, the rationale behind the approach and the specification of the objectives in this context are logical and appropriate. However, in this part of the manuscript, the authors should provide readers with much more information about the research instrument being validated – wording of items, authorship (based on the subsection Instrument, we can only guess that it was constructed by Gavilán), why was this instrument selected for validation, why was it validated on a sample of teachers, etc.

3)     Results and Discussion - This part of the study needs to be improved as well. More explanations and interpretations are needed, the results of the study should be compared with relevant previous studies, and its results should be analysed completely taking into account the specifics of schools as organisations. In addition, more connections should be included in the literature review part. The listed theoretical and practical implications are general and vague.

4)     It is not clear which part of the submitted article “suggests that implementing artificial intelligence (AI) in interventions designed to promote behavior change is cost-effective and widely available” (lines 333-334).

5)     The authors should avoid repeating the same information multiple times.

6)     References - Resources focused on teachers, their job satisfaction, and schools as work environments should be added.

Comments on the Quality of English Language

The authors should ask the help of native English speaking proof reader, because there are some major typo and linguistic mistakes that should be fixed.

Author Response

Dear Reviewer,

Thank you very much for your informed comments, which helped us so much in improving the manuscript. We appreciated the time you spent doing this and tried our best to address all your comments.

We hope that this revised version of the paper reaches the expected standard, worthy of publication in this journal.

A detailed list of answers to your comments and suggestions is reported below.

Many thanks for your time.

Best regards,

Reviewer 2 Report

Comments and Suggestions for Authors

This is an interesting quantitative study, and the findings from the quantitative analysis seem appropriate. The scope of the reported qualitative implications stretches too far beyond the actual research project. 

There is no evidence in the article of the kind of analysis that would support suggestions like that found in lines 307-313 concerning wellness programs, and therefore is not in the scope of this research.

Comments on the Quality of English Language

The English language usage needs to be edited thoroughly. There are a number of cases of incomplete sentences, grammatical errors, and punctuation issues. 

Author Response

(The authors gave the same response as above.)

Reviewer 3 Report

Comments and Suggestions for Authors

I thank the authors and the editor for the oportunity to review this manuscript. I will provide comments for possible improvement. 

Part 1.

c1 The introduction explains the rationale for this research, but lacks explanation on the research gap - the authors need to provide evidence for the importance of the proposed instrument, especially regarding other employee experience scales.

c2 In the introduction section the authors should also briefly describe their contribution with this research. 

Part 2.

c3 The authors need to provide an explicit definition of the employee experience - providing an overview of other researchers' definitions is not enough, as this research needs to give an uniform definition.

c4 The first sentence, lines 81 and 82, is not clear and does not make much sense currently. 

c5 Sentence on line 100 needs to be revised. 

c6 in line 120 the authors state that "there have been very few scholarly contributions" on this topic, so one wonders why is that, if this topic is so important? Pease provide explanation. 

c7 In lines 141 and 142 reference [66] is mentioned, but it is about customer experience, not empoyee experience. This needs to be eliminated as it is not relevant, as well as from Table 1. 

Part 3

c8 Sentence on lines 156 and 157, part 3.1, makes no sense. This part also needs to be more elaborated. 

c9 THe sentence on line 161 is hard to comprehend, please revise. 

c10 Please state if the online questionnaire was anonymous, and if so, how was that achieved?

c11 In part 3.3 the authors need to explain why did they choose the three scales already available. 

c12 In part 3.3 the authors need to explain also why did they decide to propose these three scales as dimensions of employee experience. This is crucial for this manuscript and should be elaborated extensively. 

c13 CFA should not be performed on the same dataset/sample as EFA. This is not acceptable. The authors should have obtained another data set for this, as confirmatory analysis cannot be meaningfully performed on the same data which served as input for the model. This flaw of the manuscript makes it not suitable for publication, so I will refrain from commenting discussion. I suggest that the authors conduct another round of data collection and perform CFA on that sample. 

c14 Throughout the manuscript there are numerous gramar errors. Once other flaws are mended, the manuscript needs to go through the proofreading before being considered for acceptance. 

Comments on the Quality of English Language

Throughout the manuscript there are numerous gramar errors. Once other flaws are mended, the manuscript needs to go through the proofreading before being considered for acceptance. 

Author Response

(The authors gave the same response as above.)

Round 2

Reviewer 1 Report

Comments and Suggestions for Authors

Thank you for the opportunity to review this manuscript again. I really appreciate that the authors have addressed all my comments and recommendations, have significantly improved the paper and thus increased its quality. However, I recommentd to further improve the Literature review and Discussion sections. In the Literature review (e.g. lines 116-125) business entities are paid attention to, but the specifics of school environments are still not mentioned, which to some extent also applies to the Discussion section (lines 365-377). Furthermore, in the Discussion section (lines 385-414), more connections should be included with the results of already published studies.

Comments on the Quality of English Language

There are only some minor linguistic problems in the manuscript.

Author Response

(The authors gave the same response as above.)

Reviewer 2 Report

Comments and Suggestions for Authors

Conclusions are much better supported with a clear purpose, framework, and outcomes. 

Comments on the Quality of English Language

This is acceptable. There are a few places that could flow better, but not significant enough to delay publication. 

Author Response

Dear Reviewer,
Thank you very much for your informed comments, which helped us greatly to improve the manuscript. We appreciate the time you spent on this and we did our best to respond to all your comments.
We are very pleased that we have managed to bring the latest revised version of the article up to the expected level and worthy of publication in this journal.

Thank you very much for your time.

Sincerely,

Reviewer 3 Report

Comments and Suggestions for Authors

I see significant improvements in this paper, and I have no other crucial issues. 

Comments on the Quality of English Language

English language is satisfactorily used. 

Author Response

(The authors gave the same response as above.)
